# Concentration Dependent Single Chain Properties of Poly(sodium 4-styrenesulfonate) Subjected to Aromatic Interactions with Chlorpheniramine Maleate Studied by Diafiltration and Synchrotron-SAXS

**DOI:** 10.3390/polym13203563

**Published:** 2021-10-15

**Authors:** Felipe Orozco, Thomas Hoffmann, Mario E. Flores, Judit G. Lisoni, José Roberto Vega-Baudrit, Ignacio Moreno-Villoslada

**Affiliations:** 1Laboratorio Nacional de Nanotecnología LANOTEC-CENAT, Centro Nacional de Alta Tecnología, Pavas, San José 1174-1200, Costa Rica; f.orozco.gutierrez@rug.nl (F.O.); jvegab@gmail.com (J.R.V.-B.); 2Laboratorio de Polímeros, Instituto de Ciencias Químicas, Facultad de Ciencias, Universidad Austral de Chile, Casilla 567, Valdivia 5090000, Chile; thomas.hoffmann@uach.cl (T.H.); mario.flores@uach.cl (M.E.F.); 3Facultad de Ciencias, Instituto de Ciencias Físicas y Matemáticas, Universidad Austral de Chile, Valdivia 5090000, Chile; judit.lisoni@uach.cl

**Keywords:** diafiltration, SAXS, aromatic interactions, poly(sodium 4-styrenesulfonate), chlorpheniramine, polyelectrolyte, aggregation

## Abstract

The polyelectrolyte poly(sodium 4-styrenesulfonate) undergoes aromatic–aromatic interaction with the drug chlorpheniramine, which acts as an aromatic counterion. In this work, we show that an increase in the concentration in the dilute and semidilute regimes of a complex polyelectrolyte/drug 2:1 produces the increasing confinement of the drug in hydrophobic domains, with implications in single chain thermodynamic behavior. Diafiltration analysis at polymer concentrations between 0.5 and 2.5 mM show an increase in the fraction of the aromatic counterion irreversibly bound to the polyelectrolyte, as well as a decrease in the electrostatic reversible interaction forces with the remaining fraction of drug molecules as the total concentration of the system increases. Synchrotron-SAXS results performed in the semidilute regimes show a fractal chain conformation pattern with a fractal dimension of 1.7, similar to uncharged polymers. Interestingly, static and fractal correlation lengths increase with increasing complex concentration, due to the increase in the amount of the confined drug. Nanoprecipitates are found in the range of 30–40 mM, and macroprecipitates are found at a higher system concentration. A model of molecular complexation between the two species is proposed as the total concentration increases, which involves ion pair formation and aggregation, producing increasingly confined aromatic counterions in hydrophobic domains, as well as a decreasing number of charged polymer segments at the hydrophobic/hydrophilic interphase. All of these features are of pivotal importance to the general knowledge of polyelectrolytes, with implications both in fundamental knowledge and potential technological applications considering aromatic-aromatic binding between aromatic polyelectrolytes and aromatic counterions, such as in the production of pharmaceutical formulations.

## 1. Introduction

During the last decades, we have studied the interactions between aromatic polyelectrolytes, such as poly(sodium 4-styrenesulfonate) (PSS), and low molecular-weight aromatic species (LMWS) acting as counterions, among which we can find xanthene dyes [1,2,3,4,5,6,7], redox-active tetrazolium salts [8,9,10,11], and different drugs [12,13,14,15,16]. Both complementary charged species bearing aromatic groups undergo secondary aromatic-aromatic interactions, additional to primary long-range electrostatic interactions, thus producing a reinforcement of the overall interaction. Contrary to the picture given by Manning’s counterion condensation theory [17,18,19,20,21], in which the territorial binding of counterions to polyelectrolyte chains occurs, aromatic counterions and polymeric aromatic groups produce site-specific binding, losing water molecules from their respective hydration spheres, as deduced by 1D and 2D ^1^H-NMR spectroscopies [3,4,5,6,8,13,16]. Verification of the nuclear Overhauser effect allowed for the demonstration that the interacting species approach each other by less than 5 Å. Another technique that allowed us to obtain information about the interaction between aromatic polyelectrolytes and low molecular-weight aromatic counterions was diafiltration (DF). This technique is a separation technique, which allowed the direct determination of the counterions bound to the polyelectrolyte in every instant, showing comparatively higher binding and resistance to the cleaving effect of added electrolytes in solution when contrasted with systems that do not undergo aromatic-aromatic interactions [6,12,13]. As a consequence of this interaction pattern, some properties of both the counterions and the polymers change, such as aggregation, acid-base, redox, and luminescent properties [2,4,10,13]. These interactions have also served to produce interesting higher order structures [11,22,23,24,25], and confer different properties to materials [26,27,28,29,30,31]. In particular, homogenously-dispersed photosensitizers and dyes with a controllable state of aggregation have been included in solid and semisolid materials by means of complexation with an aromatic polyelectrolyte [7,26,27]; nanoparticles of redox-active and acid-based reactive aromatic molecules have been produced in the presence of aromatic polyelectrolytes and included in solid and semisolid materials used as sensors [23,28,30].

Drug vehiculization and controlled release in matrices and nanoparticles based on aromatic-aromatic interactions have also been developed [15,16,23]. Importantly, outstanding drug loading of around 50% has been achieved, since the drug acts both as a bioactive molecule carried by the nanoparticle and as a main constituent of the carrier [15,16]. The mechanism for nanocarrier formation involving the dual function of the drug has been rationalized as the consequence of ion pair formation between the charged aromatic drug and the complementary charged polymeric aromatic residues through short-range aromatic-aromatic interactions. The occurrence of aromatic-aromatic interactions between the drug chlorpheniramine maleate (CPM) and the polyelectrolyte PSS has been reported in this context [12,13,14,16]. It was found that the extent of binding and the aggregation state of the complexes depend on the absolute and the relative concentration of the reactants. At a PSS concentration of 2 mM (in sulfonate groups per liter) DF showed drug binding of around 80% in a mixture of PSS/CPM at a sulfonate/drug stoichiometry 2:1 [12,14], forming clear solutions of non-aggregated complexes. On the contrary, at a PSS/CPM stoichiometry 2:3 and 5 mM of the polymer, higher binding, and the formation of nanoparticles were observed [16].

Ion pair formation between both charged aromatic species should imply drastic changes on chain properties in rigid polymers such as PSS. The rigidity of this polymer is due to both electrostatic repulsions between charged groups and the high volume of the aromatic rings, inducing an extended helical conformation of the polymer chain [32,33]. Chain properties of PSS have long been studied by SAXS and SANS in the presence of different salts and at several concentrations. Generally, a typical polyelectrolyte peak appears in scattering profiles, whose position depends on the concentration and nature of the counterions [34,35,36,37,38]. However, there are cases in which this typical peak does not appear, related with a high screening of electrostatic repulsions [37,38,39,40]. The effect of solvents or sulfonation degree on poly(styrene-co-styrenesulfonate) copolymers has also been studied by SANS and SAXS [41]. SANS and SAXS have been successfully used for the analysis of surfactants, colloids, powders, emulsions, nanocomposites, polymers, and macromolecules in general [42,43,44,45,46], and they offer complementary information to NMR, viscosimetry [47,48,49], conductimetry [50], and electron microscopies. It is worth mentioning the use of these techniques in complex electron-conductive system based on PSS and poly(3,4-ethylene dioxythiophene) (PEDOT), (PEDOT:PSS), whose chain properties and crystallinity are influenced by the solvent [51,52,53]. However, despite the different systems containing polymers, whose conformation properties in solution have been studied, there is no report in the literature, to the best of our knowledge, concerning the behavior of aromatic polyelectrolyte chains subjected to aromatic-aromatic interactions with aromatic low molecular-weight counterions as a function of the concentration.

In this work, we study the binding, aggregation, and chain properties in the system PSS/CPM at a sulfonate/drug stoichiometry 2:1 as a function of the system concentration in the dilute and semidilute regimes (crossover concentration between 10^−3^ and 10^−2^ M (in monomeric units) for PSS) [54,55]. DF results display novel and important features for this analytical tool for analyzing the binding of the drug to the polyelectrolyte. Synchrotron-SAXS and Dynamic Light Scattering (DLS) are used as complementary techniques to determine single correlation length chain parameters and the aggregation behavior of the system, respectively. Based on these results, we highlight a model picture for the binding and physicochemical behavior of these aromatic polyelectrolyte-aromatic counterion systems.

## 2. Theory

### 2.1. Diafiltration

Initially conceived as a separation technique for practical purposes [56,57,58,59], DF has served to calculate the thermodynamic and kinetic parameters of water-soluble polymers (WSP)/low molecular-weight species (LMWS) complexes after the development of a mathematical model to justify the DF profiles [2,3,5,8,12,60,61,62]. Thus, DF allowed the direct measurement of binding constants between WSP and LMWS, such as aromatic polyelectrolytes and aromatic counterions, providing the measurement of the stabilization effect associated to aromatic-aromatic interactions. A typical DF system is shown in Figure 1. The DF cell containing an aqueous solution of the WSP and the counterions of interest has, at the input, incoming water, and, at the output, a membrane only permeable to the LMWS. Therefore, as DF proceeds, the WSP is washed while the volume in the cell is kept constant. The filtered aqueous LMWS is collected in fractions, which are then quantified to obtain a DF profile as the plot of the natural logarithm of the concentration of the LMWS in the collected DF fractions (*ln<c_LMWS_^filtrate^>)* versus the filtration factor (*F*), defined as the ratio between the accumulative filtrate volume and the constant volume in the DF cell.

Several assumptions are made regarding the interactions between the LMWS and the WSP towards the disclosure of the information concealed in the DF profiles. (1) The total amount of LMWS is distributed in three different populations, namely free in solution, reversibly bound to the WSP (and/or to other components in the DF system), and irreversibly bound to the WSP (and/or to other components in the DF system) (see Figure 1). (2) Fast equilibrium is established between the reversibly bound fraction and the fraction free in the solution, so that the steady state approximation can be applied during filtration. (3) Interactions with the DF cell components, including the membrane, are additive to those with the WSP, so that experiments made in the absence of the WSP serve as control. Given these assumptions, a mathematical model fully described in the literature was applied to the DF profiles in order to obtain the information shown below [61,62,63].

The absolute value of the slope of the DF profile in the absence of the WSP (*k^m^*) is related to the strength of the reversible interactions between the LMWS and the DF system components. Thus, an apparent dissociation constant between the LMWS and the DF system (*K_diss_^LMWS/DS^*) can be defined and calculated as shown in Equations (1) and (2), respectively, where *c_LMWS_^free^* is the concentration of LMWS free in solution, and *c_LMWS_^rev-bound-DF^* the concentration of LMWS reversibly bound to the DF system components.
(1)$$K_{diss}^{LMWS/DS}=\frac{C_{LMWS}^{free}}{C_{LMWS}^{rev-bound-DS}}$$
(2)$$K_{diss}^{LMWS/DS}=\frac{k^{m}}{1-k^{m}}$$

Similarly, the absolute value of the slope in the presence of the WSP (*j*) is related with the strength of the reversible interactions between the LMWS and both the WSP and the DF system components. Thus, an apparent dissociation constant between the LMWS and the WSP (*K_diss_^LMWS/WSP^*), defined in Equation (3), where *c_LMWS_^rev-bound-WSP^* is the concentration of LMWS reversibly bound to the WSP, can be calculated by applying Equation (4) [62].
(3)$$K_{diss}^{LMWS/WSP}=\frac{C_{LMWS}^{free}}{C_{LMWS}^{rev-bound-WSP}}$$
(4)$$\frac{k^{m}j}{k^{m}-j}\leq K_{diss}^{LMWS/DS}\leq \frac{j}{k^{m}-j}$$

The values of *k^m^* and *j* range between 0 and 1, lower values meaning stronger interaction. Theoretically *k^m^* ≥ *j*, so that *K*_diss_^LMWS/WSP^ ranges between 0 (*j* = 0) and infinite (*j* = *k^m^*). Values of *j* = *k^m^* = 1 indicate no interaction with both the DF system components and the WSP.

In Figure 1, the LMWS referred to as irreversibly bound are the molecules that present binding processes that may be reversible with an apparent dissociation constant that tend to zero at the conditions of the experiment or show much slower equilibrium kinetics than the DF process. The fraction of LMWS that is irreversibly bound at the beginning of the DF (i.e., when *F* tends to 0) (*u*) is determined from Equation (5), where *b* is the intercept of the DF profile; *m*, the absolute value of the slope (*k^m^* or *j*); *c_LMWS_^cell-init^*, the total initial LMWS concentration; and ∆*F*, the difference in *F* value at which the filtered fractions are collected.
(5)$$u=1-\frac{b\;\;\Delta F}{C_{LMWS}^{cell-init}\left(1+e^{m∆F}\right)}$$

By subtracting the *u* value of control experiments from that of the experiments made in the presence of the WSP, the initial fraction of LMWS irreversibly bound to the WSP is obtained. Likewise, the initial fraction of LMWS that is involved in association–dissociation processes (*v*) is determined from Equation (6).
(6)v=1−u

### 2.2. SAXS

SAXS stands among the most important techniques used to analyze the conformation of polymers in solution. A simplified experimental setup is shown in Figure 2. A collimated X-ray beam impacts the sample, and the elastic component of the scattered beam is detected. The intensity pattern *I* is the fingerprint of the electron density of the sample and is a continuous function of the momentum transfer *q*, i.e., *I* = *I*(*q*). From the analysis and modelling of *I*(*q*), one can obtain the characteristic lengths, shape (including surface/volume ratio), assembling state (un/folding, aggregation, internal conformation), crystalline phases with large lattice parameters, and porosity, among other materials characteristics. In SAXS, the detection angle is far below 10°, and, depending on the wavelength of the X-ray beam, one can analyze characteristic dimensions that vary between 1 and 100 nm.

The theoretical aspects that describe *I*(*q*) are reviewed in several papers and books and the reader is directed to them for more information [36,64,65,66]. In SAXS and SANS experiments, scattering profiles may be analyzed at the very low-*q* region (*q* < 0.1 nm^−1^), where the scattering from solidlike density fluctuations is predominant, following the Guinier approximation for spherical particles:*I*(*q*) ≈ *I_G_*(0) exp[−(*R_G_*^2^*q*^2^)/3](7)
where *I_G_*(0) is the extrapolation of the intensity to *q* → 0 from the observed *q* range, and *R_G_* represents the radius of gyration of the polymeric chain, typically of some tenths of nm.

On the other hand, scattering from liquid-like or solution-like density fluctuations may be described by the Ornstein–Zernike scattering function applied in a *q*-range in both low- and high-*q* regions, where the intermolecular scattering function (the form factor) can be assumed constant [67,68], given by:*I*(*q*) = *I_OZ_*(0)/[1 + (*q**ξ*_1_)^2^](8)
where *I_OZ_*(0) is the extrapolation of the intensity to *q* → 0 from the observed *q* range, and *ξ*_1_ is the correlation length representing the static screening length (see Figure 2), corresponding to the thermal blob size. The exponent 2 is typically obtained for linear polymers in semi-dilute *θ*-solutions, adopting a random walk conformation [66]. However, a fractal exponent 1.7 (equivalent to 5/3) has been also reported to properly describe *I*(*q*) for larger domain size *ξ*_2_ that corresponds to the arrangement of the smaller domains represented by *ξ*_1_ to swallowed agglomerates (see also Figure 2) [66]. At these scale lengths, the chain conformation is a self-avoiding walk of thermal blobs. Thus, accordingly,
*I*(*q*) = *I_OZ_*(0)/[1 + (*q**ξ*_2_)^5/3^](9)

Plotting the inverse of *I*(*q*), i.e., *I*(*q*)^−1^, against *q*^2^ and *q*^5/3^ as described in Equations (10) and (11), respectively, will lead to straight lines, from which *x*_1_ and *x*_2_ can be extracted:*I*(*q*)^−1^ = *I_OZ_*(0)^−1^ + *I_OZ_*(0)^−1^*ξ*_1_^2^*q*^2^(10)
*I*(*q*)^−1^ = *I_OZ_*(0)^−1^ + *I_OZ_*(0)^−1^*ξ*_2_^5/3^*q*^5/3^(11)

## 3. Experimental Section

### 3.1. Reagents

PSS (Aldrich; M_w_ 70,000 g/mol; 206.2 g/mol of sulfonate groups, CAS No. 25704-18-1) and PAA (received from Aldrich as poly(acrylic acid) and then neutralized in aqueous solutions by adjusting the pH value to 7.5 with NaOH; M_w_ 450,000 g/mol, 72.06 g/mol of acrylic units, CAS No. 9003-01-4) were purified by DF over a regenerated cellulose membrane of a nominal molecular weight limit (NMWL) of 10,000 Da (Millipore). After the polymer solutions were washed at least eight times their initial volume, the solvent was removed by freeze-drying. CPM (Sigma, racemic mixture), NaOH (Merck), and HCl (Merck) were used as received. For all experiments and purification procedures, deionized water was used. NaOH and HCl were used to adjust the pH. The structures of the polymers and CPM are shown in Figure 3.

### 3.2. Equipment

The pH was controlled with a Thermo Fisher Scientific pHmeter (Oakton pH700, Waltham, MA, USA). Dynamic light scattering (DLS) measurements were done in a Nano ZS zetasizer equipment (Malvern, Cambridge, UK) with backscatter detection (173°), controlled by the Dispersion Technology Software (DTS 6.2, Malvern, Cambridge, UK). DF cell Amicon 8010 (10 mL capacity) with a regenerated cellulose DF membrane (Millipore) of a 5000 Da NMWL was used for DF experiments. CPM concentration in the filtration fractions was quantified using Heλios γ UV-vis spectrophotometer (Thermo Electron Corporation, Waltham, MA, USA). Synchrotron-SAXS experiments were done in the SAXS1 beamline of the Brazilian Synchrotron Light Laboratory (LNLS) in Campinas, Brazil (Full details of the SAXS line (6 October 2021) are found in https://www.lnls.cnpem.br/facilities/saxs1-en/ accessed on 31 August 2021.

### 3.3. Procedures

#### 3.3.1. Sample Preparation

WSP/CPM aqueous solutions with 2:1 molar ratio (WSP_n_/CPM_n/2_, *n* being the polymer concentration in mmol of sulfonate groups per liter (mM)) were prepared at pH 7.5, and a different total system concentration, with *n* ranging from 0.25 to 60 mM. A set of turbid suspensions of PSS_n_/CPM_n/2_ complex obtained at PSS concentration of 35, 40, and 50 mM were analyzed by DLS at 25 °C in triplicate. The hydrodynamic diameter and zeta potential values of the formed particles were considered valid under the criteria of the DTS 6.2 software (Malvern, Cambridge, UK); correlograms of the analyses is shown below.

#### 3.3.2. Diafiltration Measurements

A volume of WSP_n_/CPM_n/2_ aqueous mixtures (10 mL), with *n* ranging from 0.25 to 1.5 mM, were placed in a 10 mL DF cell bearing a 5000 Da NMWL membrane. The pH in the reservoir was also adjusted to 7.5. The experiments were performed at room temperature. During the experiment, the volume of the solution (10 mL) and pressure (3 bar) in the cell were kept constant. Fractions of 5 mL of the filtered solution were collected and the CPM concentration was quantified by UV-vis spectroscopy. DF control experiments were done in the absence of the WSP to analyze the interaction with the cell components. All experiments were carried out at least in duplicate. The results are expressed as a mean value, and their uncertainty as the standard deviation. The significance of the correlation of the independent variables *u* and *j* (and thus *K_diss_^CPM/WSP^*) was evaluated by the Pearson correlation coefficient method applied to the experimental data [69].

#### 3.3.3. Synchrotron-SAXS Measurements

The above prepared PSS_n_/CPM_n/2_ aqueous mixtures were injected in the in-vacuum liquid cell available on the beamline, consisting of two mica windows enclosing the solution with 1 mm X-ray pathlength. The total sample volume was 500 µL and the measurements were carried out at room temperature. The beamline energy was set at 8 keV, the sample to detector distance was 3 m, resulting in a *q* range spanning from 0.04 to 1.2 nm^−1^. The total acquisition time was 1000 s, transmission was corrected, and background was subtracted from all data. Data fitting was done using the free software Python Spyder3. The *q* domains that satisfy Equations (10) and (11) were searched in order to calculate *ξ*_1_ and *ξ*_2_.

## 4. Results and Discussion

### 4.1. Sample Preparation and DLS Characterization

Several samples were prepared with a stoichiometry WSP_n_/CPM_n/2_, and different values of *n*. Samples presenting PSS concentration in the range of 0.5–30 mM resulted in clear solutions. Samples presenting PSS concentration in the range of 40–60 mM precipitated. Between 30 and 40 mM nanoaggregates were found. This did not occur when PAA (pure or with CPM) or pure PSS was used. Figure 4 shows the correlograms of the DLS analyses of the samples PSS_35_/CPM_18_, PSS_40_/CPM_20_, and PSS_50_/CPM_25_. It can be seen that only the sample PSS_35_/CPM_18_ shows a steady decay on the correlation function. A hydrodynamic diameter of 322 ± 11 nm was obtained, with polydispersity index of 0.275. The zeta potential of the particles took a value of −30.90 ± 2.25 mV, high enough in absolute value to ensure stability of the aggregate. On the contrary, large, polydisperse particles were visible by the naked eye in the samples PSS_40_/CPM_20_ and PSS_50_/CPM_25_, which produced the shoulders and noisy correlograms at high correlation time values. For the PSS chain, entanglement is reported to occur beyond 100 mM for salt-free PSS (M_w_ ~ 100,000 g/mol) solutions, without undergoing precipitation [44]. Thus, it can be concluded that the presence of CPM and the occurrence of aromatic-aromatic interactions between the drug and PSS enhances polymer aggregation and system collapse in this concentration regime.

### 4.2. Diafiltration Analysis

We performed DF experiments for PSS_n_/CPM_n/2_ and PAA_n_/CPM_n/2_ samples in the dilute regime, *n* between 0.5 and 2.5 mM. The corresponding DF profiles are shown in Figure 5, and the corresponding DF parameters are listed in Table 1. All the DF profiles show good linearity, with values of *R*^2^ ≥ 0.98. At first sight, it is evident that PSS present much stronger interactions with CPM than PAA. The strength of the reversible interaction is given by the slopes of the profiles, whereas the ordinate at the origin is related with the *u* value, i.e., with the initial fraction of molecules irreversibly bound to the polymer. The difference between the two polymers regarding the strength of the interaction with CPM stands on the ability of PSS to undergo aromatic-aromatic interactions with the LMWS.

The DF parameters *v*, *u, k^m^*, *j*, *K_diss_^LMWS/DS^*, and *K_diss_^LMWS/WSP^* listed in Table 1 show, for blank experiments, *u* values very close to zero and *k_m_* values in the range of 0.79–0.86, indicating that there is no CPM irreversibly bound to the cell components, and that weak reversible interactions occur with the system components, with apparent dissociation constants (*K**_diss_^CPM/DS^*) higher than 3.8. In the case of PAA_n_/CPM_n/2_ mixtures, low *u* values are also found, ranging from 0.05 to 0.14, as well as relatively high *j* values, ranging between 0.59 and 0.69, also indicating weak interaction forces, with *K**_diss_^CPM/PAA^* higher than 2.5. On the contrary, for the PSS_n_/CPM_n/2_ mixtures, relatively high *u* values are found, ranging between 0.59 and 0.76, indicating that a significant initial fraction of the drug is irreversibly confined in the polymer domain. In addition, the fraction subjected to reversible binding presented *K**_diss_^CPM/PSS^* ranging between 0.48 and 0.78, related with *j* values ranging between 0.28 and 0.38, showing that the fraction of molecules in equilibrium that are bound to the polyelectrolyte is significantly higher than that of molecules free in solution.

It can be seen that, as the concentration of the PSS_n_/CPM_n/2_ system increases, *u* takes higher values (Figure 6a), indicating that a higher fraction of the total initial CPM molecules is irreversibly bound to the polymer at higher total concentration. A similar effect is found for the system PAA_n_/CPM_n/2_, in the low range of *u* values, presenting a smaller growth and higher relative standard deviations. The values of *j,* along with *K_diss_^CPM/WSP^*, also significantly increase as *n* increases, revealing that the fraction of reversible bound molecules, in addition to decreasing with respect to the irreversibly bound fraction, is less tightly bound to the polymer (Figure 6b,c). On the contrary, the data obtained for the PAA_n_/CPM_n/2_ system present considerable standard deviations, which prevent concluding a tendency for these two parameters.

These findings represent an interesting novelty in the development of DF as an analytical technique. The mathematical analysis of the DF profiles does not anticipate a direct physical correlation between *j* (or *K_diss_^LMWS/WSP^*) and *u*, i.e., between the strength of the reversible interactions and the fraction of molecules irreversibly bound to the polymer. However, a definite correlation between *u* and *j* (and *K_diss_^CPM/PSS^*) values for the PSS_n_/CPM_n/2_ system is found. Indeed, a linear dependency of *j* (and *K_diss_^CPM/PSS^*) with *u* is found with good linear regression factors in the range of concentration studied, as can be seen in Figure 6d. Pearson correlation coefficients of over 0.94 indicate a statistically significant linear positive correlation for both cases [69]. These results indicate that, for this system, the magnitudes represented by *u* and *j*, and thus *K_diss_^CPM/PSS^*, are physically linked, so that their values are directly correlated through the PSS_n_/CPM_n/2_ mixture’s initial concentration.

### 4.3. SAXS Analysis

Figure 7A shows SAXS results of the experimental scattering intensity *I*(*q*) as a function of the modulus of the momentum transfer vector *q* for five distinctive PSS_n_/CPM_n/2_ concentrations, with *n* ranging from 0.5 to 60 mM.

It can be seen in Figure 7A that the typical polyelectrolyte peak of PSS is not present in the PSS_n_/CPM_n/2_ complexes. The first two plots *a* and *b* correspond to low concentrated samples. The scattering of sample *c*, corresponding to PSS_10_/CPM_5.0_, yet in the typical concentration range at which many studies are reported in the literature [37,38], is significantly more intense. Sample *d*, PSS_35_/CPM_18_, shows in DLS a scattering pattern that is consistent with the formation of colloidal particles of nanometric size (around 300 nm, see Figure 4). These new conglomerates pop out in the SAXS profile as a small shoulder beginning at *q* ~ 0.06 nm^−1^. The shoulder is more clearly observed in sample *e*, PSS_60_/CPM_30_, corresponding to a system concentration at which the polymeric complexes display macroprecipitation.

The total scattering function has a positive component related with intrachain interactions and a negative component related with repulsive interchain interactions [38]. The disappearance of the polyelectrolyte peak for PSS in the presence of a large excess of NaCl or other metal counterions is explained by an increase in the compressibility of the polymeric chains and fluctuations of the interparticle distances which rises the intensity in the low-*q* region, and the increase in the fluctuations of the intersegmental distance, increasing the scattering intensity in the high-*q* region [38,70]. These effects have also been observed in the presence of divalent metal counterions where electrostatic attraction is stronger and the screening more intense [34,37,71,72]. The screening of electrostatic repulsive forces producing polymeric systems of neutral-like behavior is invoked, then, to explain the polyelectrolyte peak disappearance [38,72,73]. An interesting theoretical study analyzing expected SAXS profiles for different systems as a function of the form factor and the Bjerrum length has been reported [74]. Scattering profiles similar to those reported here are shown for polyelectrolyte systems bearing relatively high Bjerrum length, corresponding to sausage single chain conformations, provided that interchain interactions are considered negligible. However, models considering attractive interchain interactions and clustering have also been reported to be consistent with fluctuating transient aggregates that could fit to the SAXS profiles reported in this work [34,35,37,73,75]. Similar scattering profiles can be also found for rigid polyelectrolytes such as DNA [72], chondroitin sulfate, hyaluronate, or poly(aspartate) [68], proteins [76], coacervate interpolymer complexes [40,77], and even nonionic micelles formed in water [78].

Table 2 summarizes the *ξ*_1_ and *ξ*_2_ values obtained from curve fitting to Equations (10) and (11). The Ornstein-Zernike analysis shows good correlation of *I*(*q*)^−1^ vs. *q*^2^ for an extended range of data (see Figure 7B). The two more dilute samples seem to follow the signature of a Gaussian chain for a random walk conformation in a dilute environment. In addition, for samples where *n* is equal to or higher than 10, including those where nano- and macroprecipitates are observed, an also extended set of *I*(*q*)^−1^ data correlates well with *q*^5/3^ (Equation (11)), as observed in Figure 7C, showing a single polymer chain interacting equally well with itself and with the solvent producing self-avoiding walk conformations, characteristic of a fully swollen coil (Figure 2).

Figure 8 summarizes the correlation lengths obtained for the whole set of samples. The primary smaller thermal blobs showed static correlation lengths *ξ*_1_ in the range of 0.5–1.5 nm, growing monotonously with the total concentration of the system. On the other hand, the secondary larger domains showed fractal correlation lengths *ξ*_2_ in the range of 1.0–4.0 nm, also growing monotonously with the total concentration of the system, showing a larger rate, as compared to the primary blobs. This behavior is outstanding since the increase in the concentration of the system normally produces shrinking of the polymer chains and a decrease in the correlation lengths [68,70].

### 4.4. Aromatic WSP/Aromatic Counterion Complexation and Aggregation Model

At the concentration range of the experiments shown here for DF and synchrotron-SAXS, pure PSS does not form aggregates [44]. However, the occurrence of site-specific aromatic-aromatic interactions between CPM and the benzenesulfonate groups of PSS produces the decrease in the effective charge density of the polymer chains favoring intrachain attractive interactions and decreasing interchain repulsions, increasing the tendency of the macromolecule to fold. To explain the results shown in this paper, we should invoke the short-range character of aromatic-aromatic interactions. This involves the release of water from the hydration sphere of CPM and polymeric benzenesulfonate groups upon binding, producing ion pair formation. These ion pairs show a tendency to aggregate in hydrophobic domains. As depicted in Figure 9, these hydrophobic domains, composed of ion pairs and polymeric backbone folds and bundles, although transient, should contain the irreversibly bound fraction of CPM observed by DF and essentially determine the size of the thermal blobs related to the static screening length *ξ*_1_. The remaining charged hydrated polymeric segments provide charge for the system stabilization in water and the reversible interaction with the remaining fraction of the LMWS.

The confinement of CPM in hydrophobic domains increases with the system concentration, which should enhance both the compressibility of the system and intersegmental interactions [79]. The correlation between *u* and *j* in DF experiments indicates that polymer chains fold and ion pairs aggregate in hydrophobic domains, confining a higher number of CPM molecules in polymeric blobs (increasing the value of the *u* parameter). As more benzenesulfonate/CPM ion pairs are confined in hydrophobic domains, the net charge of the polymeric particles decreases, decreasing the strength of the interaction with the non-confined fraction of the LMWS (increasing the value of the *j* parameter). Together with an increase in the system concentration, an arrangement of the polymeric chain containing the thermal blobs into swollen agglomerates represented by the characteristic length *ξ*_2_ occurs. It is interesting to note that both *ξ*_1_ and *ξ*_2_ do increase with the total concentration. Short-range aromatic-aromatic interaction with the drug CPM should influence the size and mobility of the PSS segments. The thermal blobs, involving a higher number of unhydrated ion pairs stabilized in hydrophobic domains as the concentration increases, grow, and it is probable that they form clusters due to a certain tendency of CPM to self-aggregate [80]. These facts may be responsible for the increase in the thermal blob size, which further triggers an increase in the swollen fractal blobs size.

Finally, when the net charge of the particles decreases below a certain value, interchain interactions become more favorable than chain-solvent interactions, the persistence length of the electrostatic interactions is favored over the entropic effect, and the PSS_n_/CPM_n/2_ complex precipitates [44,79].

### 4.5. Final Remarks

Here, we successfully showed a statistically significant linear correlation between *j* (and *K**_diss_^CPM/PSS^*) and *u* for an aromatic WSP/aromatic LMWS system over a specific range of concentrations. In addition, we have shown the variation of the static and fractal correlation distances describing the behavior of the polymeric chains in the complex. Put together, the results shown here point at a binding and aggregation model that assumes the formation of hydrophobic domains upon the aggregation of polymeric hydrophobic segments and ion pairs formed through aromatic-aromatic interactions between the aromatic LMWS and the aromatic WSP. These separate domains, that may be considered to be two phases [35], consist of dynamic arrangements arising upon molecular interaction and aggregation: a discontinuous and transient inner hydrophobic phase, containing mainly the irreversibly bound fraction of LMWS confined in hydrophobic domains composed by ion pairs and polymeric backbone folds and bundles, and the hydrophilic phase, composed of the continuous aqueous phase. The interphase is composed by the remaining charged hydrated polymeric segments, which provides the system stabilization in water, and contains the reversibly bound counterions. Examples of two-phase aggregated systems have been described in the literature formed with PSS and cationic surfactants [39,81,82,83]. In addition to the study of polyelectrolyte/counterion interactions, DF has been shown to be a useful technique used to study interactions in nanophase-separated systems such nanodroplets of oily core stabilized by anionic and cationic surfactants interacting with the antibiotic oxytetracycline [63].

The binding model presented here, consistent with DLS, DF, and synchrotron-SAXS results, may be relevant for the interpretation of out-of-equilibrium processes in which the solvent is removed, so that the concentration of the complex increases, keeping the LMWS/WSP ratio constant. This may contribute to new knowledge involving material design and application, material properties, and functionality projection in fields such as agriculture, sensors, photocatalysts, environmental remediation, etc. Regarding drugs, the behavior of medicines based on aromatic polyelectrolytes/aromatic drug complexes interpreted under this binding model may contribute to the design of controlled drug delivery materials. In particular, it allows for the interpretation of the formation of pharmaceutical nanoformulations, in which outstanding high drug loading is achieved in nanocarriers (of around 50%), since the drug acts both as a bioactive molecule carried by the nanoparticle, and as a main constituent of the carrier [15].

## 5. Conclusions

As the concentration of the mixture PSS/CPM 2:1 in water increases in the dilute and semidilute regimes, a higher amount of CPM confines in the polymer domain. At polymer concentrations between 0.5 and 2.5 mM, the strength of the PSS_n_/CPM_n/2_ reversible interactions given by *j* (and thus *K_diss_^CPM/PSS^)* and the irreversibly bound fraction of CPM bound to PSS (*u*) are directly correlated, showing a linear tendency of positive slope (with Pearson correlation coefficients over 0.94), evidenced upon increasing the system total concentration. Thus, *u, j,* and *K_diss_^CPM/PSS^* increase along with the system total concentration. Lower affinity is found for CPM and the non-aromatic polyelectrolyte PAA, thus *j* and *K_diss_^CPM/PAA^* values showed high standard deviations, and correlations with *u* could not be found, highlighting the role of aromatic-aromatic interactions in the system behavior. Synchrotron-SAXS results display an outstanding increase in characteristic chain correlation lengths, static screening lengths *ξ*_1_ in the range 0.5–1.5 nm, and correlation lengths *ξ*_2_ in the range of 1–4 nm, following an aggregation pattern with a fractal dimension of 1.7. Nanoprecipitates of around 300 nm are found in the range of 30–40 mM, and macroprecipitates are found at a higher system concentration. A binding model has been proposed to interpret these results, so that, due to aromatic-aromatic interactions, the probability of ion pair formation between CPM and the benzene sulfonate groups of PSS increases with the total concentration of the system, as well as the probability of their aggregation. Therefore, hydrophobic domains are increasingly formed where a larger fraction of the CPM becomes irreversibly confined. Additionally, as the hydrophobic domains increase, less polyelectrolyte charged segments are available at the interface, so that the attraction of CPM molecules free in solution decreases, and the reversible interaction between the opposite charged species weakens. The increase in the drug confinement should be responsible for the increase in the static and fractal correlation lengths, observed in the increasing concentrations of the complex, and also weakens the interaction of the polymer chains with the solvent, producing precipitation at the highest concentrations evaluated. These findings contribute to the general knowledge of polyelectrolytes, with implications both in fundamental knowledge and potential technological applications considering aromatic-aromatic binding between aromatic polyelectrolytes and aromatic counterions, and, in particular, in the design of new pharmaceutical nanoformulations with outstanding high drug loading.

## Figures and Tables

**Figure 1 polymers-13-03563-f001:**
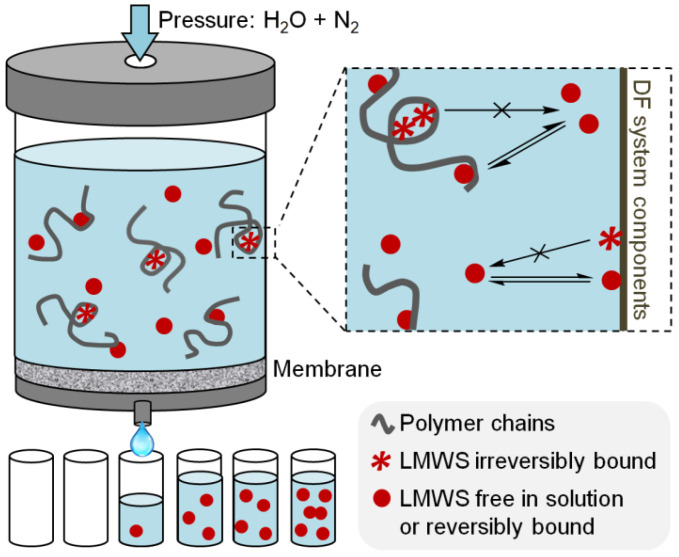
Scheme of a typical diafiltration system (**left**) and interaction model between low molecular-weight species, water-soluble polymers, and the diafiltration system components (**right**).

**Figure 2 polymers-13-03563-f002:**
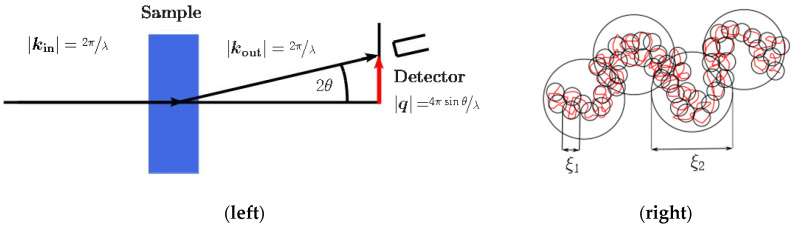
(**Left**) SAXS setting with incident and scattered wave vectors, |*k*_in_| and |*k*_out_|, respectively, and momentum transfer |*q*|; (**right**): correlation length representing the static screening length, *ξ*_1_, and fractal correlation length for larger domain size, *ξ*_2_, as determined from Equations (8) and (9).

**Figure 3 polymers-13-03563-f003:**
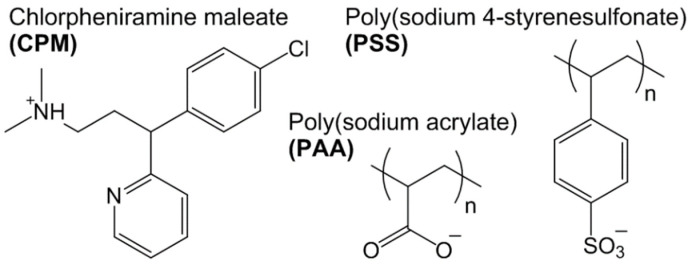
Molecular structure of CPM, PAA, and PSS.

**Figure 4 polymers-13-03563-f004:**
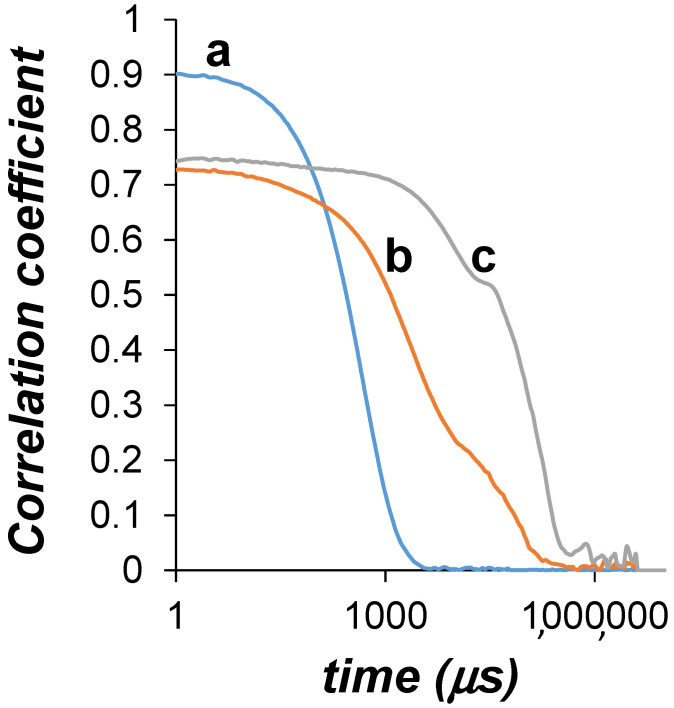
Correlograms obtained by DLS of samples PSS_35_/CPM_18_ (a), PSS_40_/CPM_20_ (b), and PSS_50_/CPM_25_ (c).

**Figure 5 polymers-13-03563-f005:**
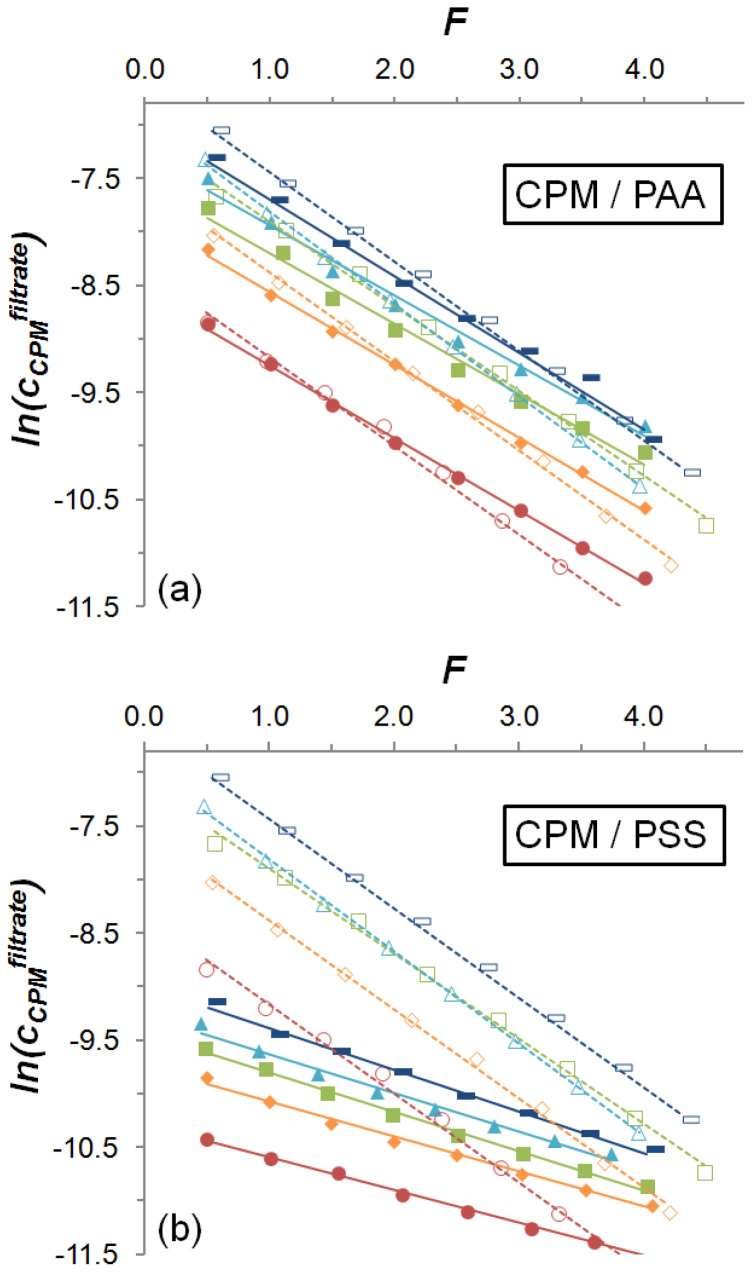
Diafiltration profiles of (**a**) PAA_n_/CPM_n/2_ and (**b**) PSS_n_/CPM_n/2_ systems at *n* = 0.50 (red circles), 1.0 (orange rhombuses), 1.5 (green squares), 2.0 (light blue triangles), and 2.5 (blue rectangles). Corresponding blank experiments made in the absence of polyelectrolytes are plotted as empty symbols.

**Figure 6 polymers-13-03563-f006:**
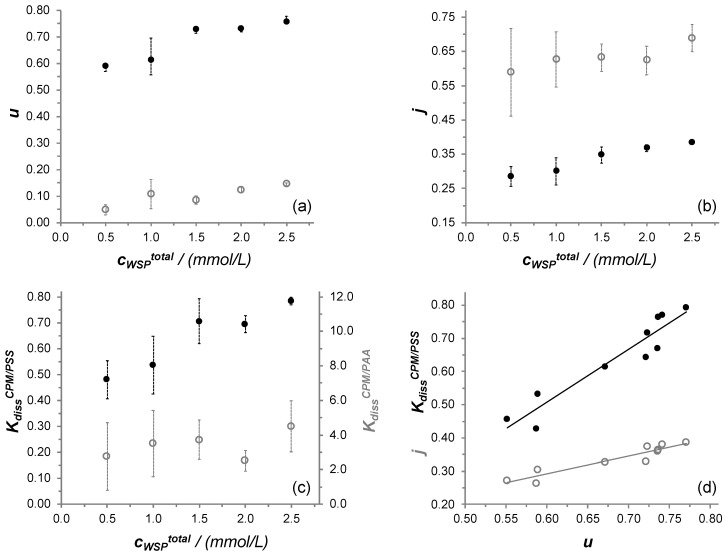
*u* values (**a**), *j* values (**b**), and *K_diss_^CPM/WSP^* values (**c**), plotted against the initial polyelectrolyte concentration (*c_WSP_^total^*) for PAA_n_/CPM_n/2_ (grey empty circles) and PSS_n_/CPM_n/2_, systems (black circles). *K_diss_^CPM/PSS^* (black circles) (y = 1.6x − 0.44; *R*^2^ = 0.90) and *j* (grey empty circles) (y = 0.53x − 0.029; *R*^2^ = 0.89) values plotted against *u* (plotting each individual experiment for all PSS_n_/CPM_n/2_ systems) (**d**).

**Figure 7 polymers-13-03563-f007:**
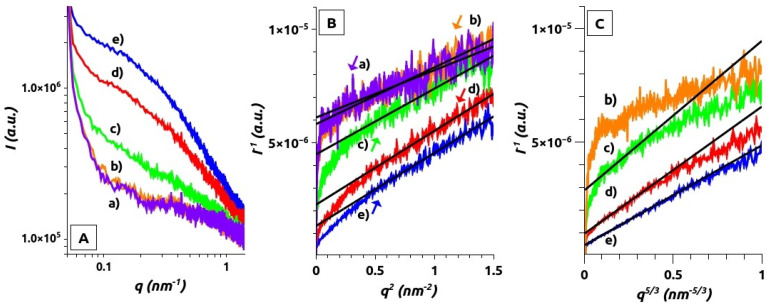
Synchrotron-SAXS results for selected PSS_n_/CPM_n/2_ concentration values: (**A**) *I*(*q*) vs. *q*; (**B**) *I*(*q*)^−1^ vs. *q*^2^, and fitted curves obtained applying Equation (10) to a set of (*I*(*q*)^−1^, *q*^2^) values; (**C**) *I*(*q*)^−1^ vs. *q*^5/3^ and fitted curves obtained applying Equation (11) to a set of (*I*(*q*)^−1^, *q*^5/3^) values. (a) *PSS_0.5_/CPM_0.25_* (mauve), (b) *PSS_2.0_/CPM_1.0_* (orange), (c) *PSS_10_/CPM_5.0_* (green), (d) *PSS_35_/CPM_18_* (red), (e) *PSS_60_/CPM_30_* (blue).

**Figure 8 polymers-13-03563-f008:**
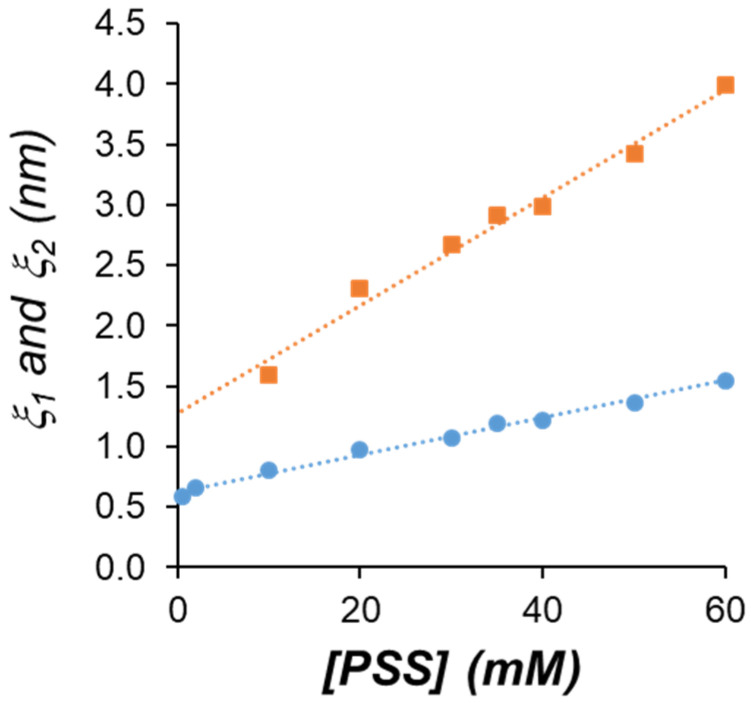
Correlation lengths *ξ*_1_ (circles) and *ξ*_2_ (squares) obtained from Equations (10) and (11), respectively. Linear regression functions are y = 0.0153 + 0.6238 (*R*^2^ = 0.99) for *ξ*_1_ vs. [PSS] and y = 0.0447 + 1.2751 (*R*^2^ = 0.98) for *ξ*_2_ vs. [PSS].

**Figure 9 polymers-13-03563-f009:**
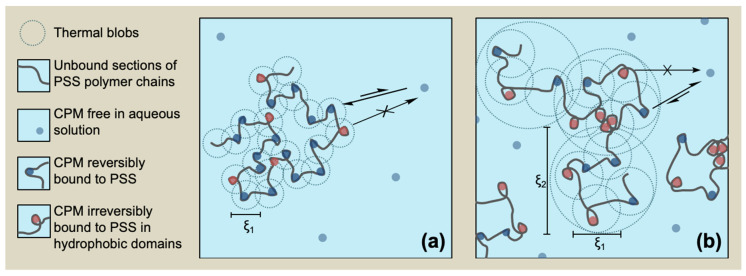
Molecular diagram of the PSS_n_/CPM_n/2_ system at different concentrations where *ξ*_1_ and *ξ*_2_, as well as the irreversibly bound fraction of CPM confined in folds and bundles of the polyelectrolyte chain increase at an increasing system concentration: (**a**) dilute regime; (**b**) semidilute regime.

**Table 1 polymers-13-03563-t001:** WSP_n_/CPM_n/2_ system formulations, the resulting DF parameters, and the linear adjustment of the DF profiles with the corresponding linear regression factors (*R*^2^).

	*c_CPM_^total^* mM	*c_WSP_^total^* mM	*v*	*u*	*j* and (*k^m^*)	*K_diss_^LMWS/WSP^* and (*K_diss_^LMWS/DS^*)	Linear Adjustment	*R* ^2^
Blank	0.25	-	1.03	−0.03	(0.83)	(4.9)	y = −0.83x − 7.6	0.99
	0.50	-	0.97	0.03	(0.81)	(4.3)	y = −0.81x − 7.6	0.99
	0.75	-	0.94	0.06	(0.79)	(3.8)	y = −0.79x − 7.1	1.00
	1.00	-	0.94	0.06	(0.86)	(6.1)	y = −0.86x − 6.9	1.00
	1.25	-	1.03	0.03	(0.84)	(5.3)	y = −0.84x − 6.6	1.00
PAA	0.25	0.5	0.95 ± 0.02	0.05 ± 0.02	0.59 ± 0.13	2.8 ± 2.0	(−0.59 ± 0.13)x + (−8.7 ± 0.2)	1.00 ± 0.00
	0.50	1.0	0.89 ± 0.05	0.11 ± 0.05	0.63 ± 0.08	3.5 ± 1.9	(−0.63 ± 0.08)x + (−8.0 ± 0.2)	0.99 ± 0.01
	0.75	1.5	0.92 ± 0.02	0.08 ± 0.02	0.63 ± 0.04	3.7 ± 1.1	(−0.63 ± 0.04)x + (−7.6 ± 0.1)	0.98 ± 0.01
	1.00	2.0	0.88 ± 0.01	0.12 ± 0.01	0.62 ± 0.04	2.5 ± 0.6	(−0.62 ± 0.04)x + (−7.4 ± 0.1)	0.99 ± 0.00
	1.25	2.5	0.86 ± 0.01	0.14 ± 0.01	0.69 ± 0.04	4.5 ± 1.5	(−0.69 ± 0.04)x + (−7.0 ± 0.1)	0.99 ± 0.00
PSS	0.25	0.5	0.41 ± 0.00	0.59 ± 0.00	0.28 ± 0.03	0.48 ± 0.07	(−0.28 ± 0.03)x + (−10.4 ± 0.1)	0.98 ± 0.02
	0.50	1.0	0.39 ± 0.08	0.61 ± 0.08	0.30 ± 0.04	0.54 ± 0.11	(−0.30 ± 0.04)x + (−9.7 ± 0.1)	0.98 ± 0.01
	0.75	1.5	0.27 ± 0.01	0.73 ± 0.01	0.35 ± 0.02	0.70 ± 0.09	(−0.35 ± 0.02)x + (−9.5 ± 0.0)	0.99 ± 0.00
	1.00	2.0	0.27 ± 0.01	0.73 ± 0.01	0.37 ± 0.01	0.69 ± 0.03	(−0.37 ± 0.10)x + (−9.2 ± 0.1)	0.98 ± 0.00
	1.25	2.5	0.24 ± 0.02	0.76 ± 0.02	0.38 ± 0.00	0.78 ± 0.01	(−0.38 ± 0.00)x + (−9.0 ± 0.1)	0.99 ± 0.01

**Table 2 polymers-13-03563-t002:** *ξ*_1_ and *ξ*_2_ correlation lengths for different PSS_n_/CPM_n/2_ system formulations.

PSS_n_/CPM_n/2_ (mM/mM)	*q* Range for Equation (10) (nm^−1^)	*R* ^2^	*ξ*_1_ ± *δ**ξ*_1_ (nm)	*q* Range for Equation (11) (nm^−1^)	*R* ^2^	*ξ*_2_ ± *δ**ξ*_2_ (nm)
0.5/0.25	0.49–1.23	0.78	0.60 ± 0.01	-	-	-
2.0/1.0	0.53–1.27	0.90	0.70 ± 0.01	-	-	-
10/5.0	0.61–1.22	0.90	0.80 ± 0.01	0.20–0.52	0.924	1.70 ± 0.02
20/10	0.67–1.19	0.94	1.00 ± 0.01	0.22–0.53	0.978	2.30 ± 0.03
30/15	0.74–1.21	0.85	1.10 ± 0.02	0.20–0.54	0.976	2.70 ± 0.05
35/18	0.74–1.21	0.95	1.20 ± 0.02	0.20–0.55	0.989	2.90 ± 0.03
40/20	0.73–1.20	0.94	1.20 ± 0.02	0.20–0.55	0.992	3.00 ± 0.04
50/25	0.73–1.20	0.97	1.40 ± 0.02	0.20–0.55	0.993	3.40 ± 0.03
60/30	0.67–1.17	0.98	1.50 ± 0.02	0.18–0.53	0.995	4.0 ± 0.1

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
