# Peer review of "Concentration Dependent Single Chain Properties of Poly(sodium 4-styrenesulfonate) Subjected to Aromatic Interactions with Chlorpheniramine Maleate Studied by Diafiltration and Synchrotron-SAXS"

_polymers, 2021, doi:10.3390/polym13203563_

Round 1

Reviewer 1 Report

The manuscript entitled "Concentration Dependent Single Chain Properties of Poly(So-dium 4-Styrenesulfonate) subjected to Aromatic Interactions with Chlorpheniramine Maleate Studied by Diafiltration and Synchrotron-SAXS" presented by Orosco et al is well-written, results are well-presented but, in my opnion, needs further revision to be considered for publication in "Polymers".

In the following lines, you will see my queries about the manuscript in the present form:

Q1)

In the "Introduction" section (section 1, page 2), quote: "...However, despite the different systems containing polymers whose conformation properties in solution have been studied, there is no report in the literature, to the best of our knowledge, concerning the behavior...".

Here, the specific state-of-the-art is rather vague for SAXS experiments on similar polymeric materials in the solid state or in solution. Authors can enrich this section by refering to:
-J. Mater. Chem., 2011,21, 1607-1614.
-Macromolecules 2012, 45, 9, 3859–3865.
-Chemical Physics Letters 2006, 425, 1–3, 114-117.
-Macromolecules 2010, 43, 23, 9881–9891.
-Polymer Journal 2014,  46, 155–159. 
-Macromolecules 2009, 42, 24, 9568–9580.
-J. Phys. Chem. B 2015, 119, 34, 11010–11018.

Q2) In the "Theory" section (section 2, page 5), please revise the equations. For instance, I note that Guinier equation is not correct, as it is proportional to the exponent and not the inverse as shown in Eq.7. Also, authors must define the q range where Eqs. 10 and 11 are valid, at least generically for low-q and high-q regions.

Q3) In the "Theory" section (section 2, page 5), quote: "...However, a fractal exponent 1.7 (equivalent to 5/3) has been also reported to properly describe I(q) for larger domain size ...". 

Here, besides reference 51, can authors estimate their own fractal exponent performing the analysis at high-q region? If not, explain why.

Q4) In the "Results and discussion" section (section 4, page 11), quote: "...The summary of the RG results for samples a - e is given in Table 2. For the low concentrated solutions, no significant differences are found, resulting in RG of ~4 nm. As the concentration of the complexes increases, RG slightly increases reaching a maximum value of ~6 nm for the highest concentration..."

Here, can authors explain the relation between these Rg values and those estimated for small and large domain size reported above? An inclusion of Rg in the scheme of Figure 9 could be helpful.
Also, aren't these Rg values possibly affected by the structure factor [S(q)] in the higher concentrations, please explain further.

Author Response

Answer to Comments and Suggestions for Authors

Reviewer 1

The manuscript entitled "Concentration Dependent Single Chain Properties of Poly(So-dium 4-Styrenesulfonate) subjected to Aromatic Interactions with Chlorpheniramine Maleate Studied by Diafiltration and Synchrotron-SAXS" presented by Orosco et al is well-written, results are well-presented but, in my opnion, needs further revision to be considered for publication in "Polymers".

In the following lines, you will see my queries about the manuscript in the present form:

Comment 1.1. In the "Introduction" section (section 1, page 2), quote: "...However, despite the different systems containing polymers whose conformation properties in solution have been studied, there is no report in the literature, to the best of our knowledge, concerning the behavior...".

Here, the specific state-of-the-art is rather vague for SAXS experiments on similar polymeric materials in the solid state or in solution. Authors can enrich this section by refering to:
-J. Mater. Chem., 2011,21, 1607-1614.
-Macromolecules 2012, 45, 9, 3859–3865.
-Chemical Physics Letters 2006, 425, 1–3, 114-117.
-Macromolecules 2010, 43, 23, 9881–9891.
-Polymer Journal 2014,  46, 155–159. 
-Macromolecules 2009, 42, 24, 9568–9580.
-J. Phys. Chem. B 2015, 119, 34, 11010–11018.

Answer 1.1. All proposed references have been included in the introduction section, that has been expanded. They reinforce our statement, since none of these references reports the behavior of aromatic polyelectrolyte chains subjected to aromatic-aromatic interactions with aromatic low molecular-weight counterions as a function of the concentration. The new paragraph now reads:

“Ion pair formation between both charged aromatic species should imply drastic changes on chain properties in rigid polymers such as PSS. The rigidity of this polymer is due to both electrostatic repulsions between charged groups, and the high volume of the aromatic rings, inducing an extended helical conformation of the polymer chain [32,33]. Chain properties of PSS have long been studied by SAXS and SANS in the presence of different salts and at several concentrations. Generally, a typical polyelectrolyte peak appears in scattering profiles whose position depends on the concentration and nature of the counterions [34-38]. However, there are cases in which this typical peak does not appear, related with a high screening of electrostatic repulsions [37-40]. The effect of solvents or sulfonation degree on poly(styrene-co-styrenesulfonate) copolymers has also been studied by SANS and SAXS [41]. SANS and SAXS have been successfully used for the analysis of surfactants, colloids, powders, emulsions, nanocomposites, polymers, and macromolecules in general [42-46], and they offer complementary information to NMR, viscosimetry [47-49], conductimetry [50], and electron microscopies. It is worth to mention the use of these techniques in complex electron-conductive system based on PSS and poly(3,4-ethylene dioxythiophene) (PEDOT), (PEDOT:PSS), whose chain properties and crystallinity are influenced by the solvent [51-53]. However, despite the different systems containing polymers whose conformation properties in solution have been studied, there is no report in the literature, to the best of our knowledge, concerning the behavior of aromatic polyelectrolyte chains subjected to aromatic-aromatic interactions with aromatic low molecular-weight counterions as a function of the concentration.”

Comment 1.2. In the "Theory" section (section 2, page 5), please revise the equations. For instance, I note that Guinier equation is not correct, as it is proportional to the exponent and not the inverse as shown in Eq.7. Also, authors must define the q range where Eqs. 10 and 11 are valid, at least generically for low-q and high-q regions.

Answer 1.2. The equations have been corrected and the q range where Eqs 10 and 11 has been included.

Comment 1.3. In the "Theory" section (section 2, page 5), quote: "...However, a fractal exponent 1.7 (equivalent to 5/3) has been also reported to properly describe I(q) for larger domain size ...". 

Here, besides reference 51, can authors estimate their own fractal exponent performing the analysis at high-q region? If not, explain why.

Answer 1.3. Searching for other fractal exponents may be done for any q domain in the high- or low-q regions. However, a theory behind should be developed to interpret other fractal exponents in terms of chain conformation. In fact, what is searched is the q domain at which the fractal exponent 5/3 is found. This has been reinforced in the Experimental section, that now reads:

“Data fitting was done using the free software Python Spyder3. The q domains that satisfy Equations (10) and (11) were searched in order to calculate x1 and x2.”

Comment 1.4. In the "Results and discussion" section (section 4, page 11), quote: "...The summary of the RG results for samples a - e is given in Table 2. For the low concentrated solutions, no significant differences are found, resulting in RG of ~4 nm. As the concentration of the complexes increases, RG slightly increases reaching a maximum value of ~6 nm for the highest concentration..."

Here, can authors explain the relation between these Rg values and those estimated for small and large domain size reported above? An inclusion of Rg in the scheme of Figure 9 could be helpful.
Also, aren't these Rg values possibly affected by the structure factor [S(q)] in the higher concentrations, please explain further.

Answer 1.3. The correlation distances x1 and x2 are related to statistical blobs formed by chain segments while Rg is related with the whole chain behavior, so that, they may have the same tendency to increase with the system concentration. However, after a discussion within the research team, we have decided to remove the short discussion on Rg. The reason is that the Synchrotron in Brazil did not give enough data at very low-q range, so that the Guinier equation could not be used. Thus, we used the SasView software as an alternative to analyze Rg. However, the results were too noisy, and, although we found interesting the increase on Rg with the complex concentration, as observed with x1 and x2, associate errors in the calculation could lead to misinterpretations that we prefer to avoid at this stage.

Reviewer 2 Report

The article titled “Concentration Dependent Single Chain Properties of Poly(Sodium 4-Styrenesulfonate) subjected to Aromatic Interactions with Chlorpheniramine Maleate Studied by Diafiltration and Synchrotron-SAXS” by Orozco et al. has been reviewed. The work is properly designed, and the results are well presented where the aromatic-aromatic interactions between NaPSS and CPM have been investigated. However, addressing a few of the following concerns will further improve the quality of the work.

  1. In the introduction the authors have stated “……in a mixture of PSS / CPM at a stoichiometry 2 : 1,”; however the term ‘stoichiometry’ needs to be defined or explained so that the readers can easily understand the type/kind of ratio mentioned? Are the authors referring to equivalent weight of the polymer or the molecular weight in this ratio?
  2. The authors have mentioned, “Chain properties of PSS have long been studied by SAXS and SANS in the presence of different salts and at several concentrations”, moreover they have also been very popularly studied by exploring their ionic conductivity, viscosity, (J. Mol. Structure, 2020, 1199,126992; Macromolecules, 2021, 54(3), 1375-1387; RSC Adv., 2015,5, 54890-54898; Polym. Int. 2014, 63(11), 1959-1964), etc. Thus, authors should also mention these.
  3. In the last paragraph of Introduction section, the authors have mentioned “dilute and semidilute regimes”, which needs to be defined here for the polyelectrolyte used in this so that the readers can have clear idea about the significance of these regimes in polyelectrolyte studies in solution phase.
  4. A small introduction about the DF is to be provided into the introduction section, where the term first appears. Authors have presented the introduction and theory of DF in section 2.1. The theory presented at section 2.1 is okay, however an introduction to DF should be placed at the introduction section to make this process clear prior to the theory placed in section 2.1.
  5. The equation numbers throughout the manuscript are placed too close to the equations. These equation numbers need to be presented properly following the standard format.
  6. The figure captions should be self-explanatory, thus use of abbreviated terms should be avoided.
  7. In section 3.1, provide the product code of the three main materials, NaPSS, NaPAA and CPM, so that the materials can be easily found and the results can be reproduced, when a reader is interested.
  8. In section 3.3, the different techniques mentioned can be arranged in separate paragraphs within the same section.

Author Response

Answer to Comments and Suggestions for Authors

Reviewer 2:

The article titled “Concentration Dependent Single Chain Properties of Poly(Sodium 4-Styrenesulfonate) subjected to Aromatic Interactions with Chlorpheniramine Maleate Studied by Diafiltration and Synchrotron-SAXS” by Orozco et al. has been reviewed. The work is properly designed, and the results are well presented where the aromatic-aromatic interactions between NaPSS and CPM have been investigated. However, addressing a few of the following concerns will further improve the quality of the work.

Comment 2.1. In the introduction the authors have stated “……in a mixture of PSS / CPM at a stoichiometry 2 : 1,”; however the term ‘stoichiometry’ needs to be defined or explained so that the readers can easily understand the type/kind of ratio mentioned? Are the authors referring to equivalent weight of the polymer or the molecular weight in this ratio?

Answer 2.1: The stoichiometry has been explicitly defined by that the polymer concentration is measured in mole of sulfonate groups per liter, thus the polymer molecular weight (equivalent) is given in g of polymer / mol of sulfonate groups. Thus, in the introduction and in the experimental section, now reads:

“It was found that the extent of binding and the aggregation state of the complexes depend on the absolute and the relative concentration of the reactants. At PSS concentration of 2 mM (in sulfonate groups per liter). DF showed drug binding of around 80 % in a mixture of PSS / CPM at a sulfonate/drug stoichiometry 2 : 1 [12,14], forming clear solutions of non-aggregated complexes…

In this work we will study binding, aggregation, and chain properties in the system PSS / CPM at a sulfonate/drug stoichiometry 2 : 1 as a function of the system concentration, in the dilute and semidilute regimes (crossover concentration between 10-3 and 10-2 M (in monomeric units) for this polyelectrolyte) [54,55]…

3.1. Reagents. PSS (Aldrich; Mw 70 000 g/mol; 206.2 g/mol of sulfonate groups, CAS No. 25704-18-1) and PAA (received from Aldrich as poly(acrylic acid) and then neutralized in aqueous solutions by adjusting the pH value to 7.5 with NaOH; Mw 450 000 g/mol, 72.06 g/mol of acrylic units, CAS No. 9003-01-4)…

3.3. Procedures. Sample preparation. WSP / CPM aqueous solutions with 2 : 1 molar ratio (WSPn / CPMn/2, n being the polymer concentration in mmol of sulfonate groups per liter (mM)) were prepared at pH 7.5, and…”

Comment 2.2. The authors have mentioned, “Chain properties of PSS have long been studied by SAXS and SANS in the presence of different salts and at several concentrations”, moreover they have also been very popularly studied by exploring their ionic conductivity, viscosity, (J. Mol. Structure, 2020, 1199,126992; Macromolecules, 2021, 54(3), 1375-1387; RSC Adv., 2015,5, 54890-54898; Polym. Int. 2014, 63(11), 1959-1964), etc. Thus, authors should also mention these.

Answer 2.2: All proposed references have been included in the introduction section, that has been expanded. The text now reads:

“The effect of solvents or sulfonation degree on poly(styrene-co-styrenesulfonate) copolymers has also been studied by SANS and SAXS [41]. SANS and SAXS have been successfully used for the analysis of surfactants, colloids, powders, emulsions, nanocomposites, polymers, and macromolecules in general [42-46], and they offer complementary information to NMR, viscosimetry [47-49], conductimetry [50], and electron microscopies. It is worth to mention the use of these techniques in complex electron-conductive system based on PSS and poly(3,4-ethylene dioxythiophene) (PEDOT), (PEDOT:PSS), whose chain properties and crystallinity are influenced by the solvent [51-53]”

Comment 2.3. In the last paragraph of Introduction section, the authors have mentioned “dilute and semidilute regimes”, which needs to be defined here for the polyelectrolyte used in this so that the readers can have clear idea about the significance of these regimes in polyelectrolyte studies in solution phase.

Answer 2.3: Insights of the crossover concentration between dilute and semidilute regimes considered for this polymer has been added, together with two references. The las paragraph in the introduction now reads:

“In this work we will study binding, aggregation, and chain properties in the system PSS / CPM at a sulfonate/drug stoichiometry 2 : 1 as a function of the system concentration, in the dilute and semidilute regimes (crossover concentration between 10-3 and 10-2 M (in monomeric units) for this polyelectrolyte) [54,55]”

However, as we do not have an actual crossover concentration for the complex of PSS with the drug, both species subjected to mutual aromatic-aromatic interactions, we have avoided using the terms dilute and semidilute for specific system concentrations.

Comment 2.4. A small introduction about the DF is to be provided into the introduction section, where the term first appears. Authors have presented the introduction and theory of DF in section 2.1. The theory presented at section 2.1 is okay, however an introduction to DF should be placed at the introduction section to make this process clear prior to the theory placed in section 2.1.

Answer 2.4: A small introduction on DF has been included in the introduction section, that has now been expanded. The text now reads:

“Another technique that allowed us obtaining information about the interaction between aromatic polyelectrolytes and low molecular-weight aromatic counterions is diafiltration (DF). This technique is a separation technique, which allowed the direct determination of the counterions bound to the polyelectrolyte in every instant, showing comparative higher binding and resistance to the cleaving effect of added electrolytes in solution when contrasted to systems that do not undergo aromatic-aromatic interactions [6,12,13].”

Comment 2.5. The equation numbers throughout the manuscript are placed too close to the equations. These equation numbers need to be presented properly following the standard format.

Answer 2.5: Equation numbers have been moved to the right end of the lines, and the format checked.

Comment 2.6. The figure captions should be self-explanatory, thus use of abbreviated terms should be avoided.

Answer 2.6: The figure captions have been improved according to the referee´s directions.

Comment 2.7. In section 3.1, provide the product code of the three main materials, NaPSS, NaPAA and CPM, so that the materials can be easily found and the results can be reproduced, when a reader is interested.

Answer 2.7: The CAS No of the polymers have been included in section 3.1.

Comment 2.8. In section 3.3, the different techniques mentioned can be arranged in separate paragraphs within the same section.

Answer 2.8: Section 3.3 has been separated in paragraphs according to the referee´s directions.

Round 2

Reviewer 1 Report

Authors have addressed my queries satisfactorily, now I recommend the publication of the manuscript in its revised form.

Reviewer 2 Report

The authors have improved the manuscript substantially. It may be accepted for publication.